



# Flash drought onset over the Contiguous United States: Sensitivity of inventories and trends to quantitative definitions

Mahmoud Osman[1], Benjamin F. Zaitchik[1], Hamada S. Badr[1], Jordan I. Christian[2], Tsegaye Tadesse[3], Jason A. Otkin[4], Martha C. Anderson[5]

[1] *Department of Earth and Planetary Sciences, Johns Hopkins University, Baltimore, MD, USA*

[2] *School of Meteorology, University of Oklahoma, Norman, OK, USA*

[3] *National Drought Mitigation Center, University of Nebraska-Lincoln, NE, USA*

[4] *Space Science and Engineering Center, Cooperative Institute for Meteorological Satellite Studies, University of Wisconsin–Madison, WI, USA*

[5] *Hydrology and Remote Sensing Laboratory, Agricultural Research Service, USDA, MD, USA*

*Correspondence to:* Mahmoud Osman (mahmoud.osman@jhu.edu)

**Abstract:**

The term "flash drought" is frequently invoked to describe droughts that develop rapidly over a relatively short timescale. Despite extensive and growing research on flash drought processes, predictability, and trends, there is still no standard quantitative definition that encompasses all flash drought characteristics and pathways. Instead, diverse definitions have been proposed, supporting wide-ranging studies of flash drought but creating the potential for confusion as to what the term means and how to characterize it. Use of different definitions might also lead to different conclusions regarding flash drought frequency, predictability, and trends under climate change. In this study, we compared five previously published definitions, a newly proposed definition, and an operational satellite-based drought monitoring product to clarify conceptual differences and to investigate the sensitivity of flash drought inventories and trends to the choice of definition. Our analyses indicate that the newly introduced Soil Moisture Volatility Index definition effectively captures flash drought onset in both humid and arid regions. Analyses also showed that estimates of flash drought frequency, spatial distribution, and seasonality vary across the contiguous U.S. depending upon which definition is used. Definitions differ in their representation of some of the largest and most widely studied flash droughts of recent years. Trend analysis indicates that definitions that include air temperature show significant increases in flash droughts over the past forty years, but few trends are evident for definitions based on other surface conditions or fluxes. These results indicate that "flash drought" is a composite term that includes several types of event, and that clarity in definition is critical when monitoring, forecasting, or projecting the drought phenomenon.





## 1. Introduction:

The concept of *flash drought* (Svoboda et al., 2002) has drawn considerable attention in recent years (Anderson et al., 2013; Basara
et al., 2019; Chen et al., 2019; Christian et al., 2019a; Ford and Labosier, 2017; Gerken et al., 2018; Hunt et al., 2009; Koster et
al., 2019; Li et al., 2020; Liu et al., 2020; Otkin et al., 2013, 2018, 2019; Pendergrass et al., 2020; Yuan et al., 2019). While there
is no single quantitative definition for what constitutes such an event, it is widely understood that some of the most damaging
droughts in the United States in the past decade have been flash droughts, in that they have emerged rapidly and caused significant
damage to natural and managed vegetation (Zhang and Yuan, 2020). These flash droughts have been difficult to predict and monitor
(Chen et al., 2019; Ford and Labosier, 2017; Pendergrass et al., 2020). There is also an understanding that many flash droughts are
triggered or exacerbated by high temperatures leading to increased evaporative demand (Anderson et al., 2013; McEvoy et al.,
2016; Otkin et al., 2013, 2018). The significant impacts and limited predictability of these events and their apparent link to high
temperatures has led to studies of customized event inventories, forecast methods, and trend analysis (e.g., Mo and Lettenmaier,
2015, 2016; Ford and Labosier, 2017).
The burst of research interest in flash droughts has yielded useful insights on process and predictability. But in the absence of a
single generalizable definition, there is potential for divergent results and general fragmentation of research agendas insomuch as
the same term "flash drought" might be applied in inconsistent ways. This potential is evident in Fig. 1, which offers a simplified
schematic of flash drought processes. Different colored boxes in the figure indicate variables or processes that are included in
different published definitions of flash droughts. For example, as will be described in detail in the methods and results section, the
"heat wave flash drought" definition (Mo and Lettenmaier, 2015) stresses the role of temperature anomalies and identifies features
with short duration, while definitions based on rapid soil drying (e.g., Hunt *et al.*, 2009; Ford and Labosier, 2017; Yuan et al.,
2019) focus on the rate of change in soil moisture. Other researchers (e.g., Christian *et al.*, 2019a; Pendergrass *et al.*, 2020) have
proposed definitions that use actual and/or potential evapotranspiration anomalies, and still others have applied multivariate
products like Quick Drought Response Index (QuickDRI) hybrid satellite-based maps or the United States Drought Monitor, which
consider vegetation status and agricultural impacts in addition to hydrological variables (e.g., Chen *et al.*, 2019).
Given this range of variables used to assess flash drought risk and diagnose its occurrence, it is possible that the definitions are
capturing partially or entirely different pathways in the flash drought process (i.e., different boxes in Fig. 1).

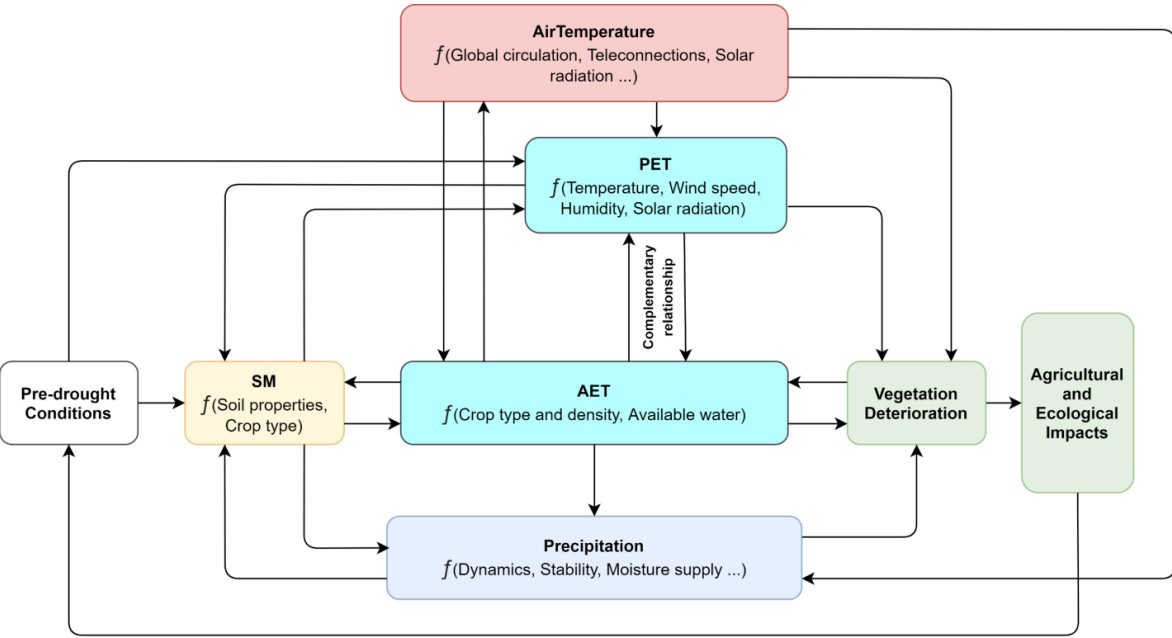


*Figure 1: Schematic of flash drought states and processes. Arrows indicate suggested feedback directions and their relation to the*
*process or variable (for simplicity, not all proposed feedbacks are represented here). Each color represents a core group of*
*processes that can be used to define the onset of flash drought event.*





This diversity of definitions is not necessarily a weakness of the literature. Flash droughts, like droughts in general, are likely a
composite class for which no single definition can meet all needs (Heim Jr., 2002). But it is important to understand the extent to
which flash drought inventories are sensitive to the choice of definition, as these inventories are the basis for assessing which
regions are most vulnerable to flash droughts and whether there are trends in flash drought frequencies in any region. These
inventories also determine the population of flash drought events used as prediction targets when developing forecast systems.
With this motivation, this study presents inventories generated using a number of prominent published flash drought definitions.
In some cases, these definitions have already been used to generate inventories, and we simply recalculate those inventories using
a common set of input data and thresholds. In other cases, the definitions were published without an inventory, and sometimes
without any recommended thresholds. For those definitions we adapt the descriptive definitions to a quantitative framework for
the purpose of creating an inventory. In addition, we propose our own definition, based on root zone soil moisture volatility, which
is designed to complement existing definitions, and we compare all proposed flash drought definitions to selected indicators of
drought impacts.
In comparing definitions, we can: (1) evaluate whether the current diversity of flash drought definitions is convergent or
divergent—i.e., is the concept of flash drought robust to different definitions; (2) identify and characterize the potential divergence
between definitions,  and assess whether different definitions capture similar processes but diverge because of threshold effects,
timing of diagnosis, or extent of drought, or whether they capture fundamentally different types of events; and (3) identify events
that are considered to be flash droughts under some definitions but not others, and learn from these case studies what elements of
a definition are important when attempting to identify particular kinds of flash droughts.
**2.   Data and Methods:**
2.1.  Flash Drought Definitions:
We inventory potential flash drought events using a range of definitions. As we are concerned primarily with drought impacts on
agriculture and natural vegetation, we focus our analysis on spring (MAM), summer (JJA) and fall (SON) and do not consider
winter months. We consider seven methods for identifying a flash drought. The first—the Soil Moisture Volatility Index (SMVI)—
is a new definition proposed here. The next five are drawn from published literature on flash droughts, and the seventh is based on
a remotely sensed product designed to be sensitive to rapid onset droughts. Where data coverage allows, we use the 1979-2018
period for index calculation and comparisons. For some products, there is a more limited data record, and in those cases, we use
all available data. Differences in input dataset requirements and baseline period can affect comparisons across definition, and are
noted when relevant. Here we describe each definition and present the datasets used to calculate them.
1.   SMVI (Soil Moisture Volatility Index):
As flash droughts are characterized by rapid onset, we adopt an approach inspired by studies of market volatility, where robust
identification of rapid yet significant changes in stock prices is critical. In this definition, a flash drought is said to occur when: (1)
the 1-pentad (5 day) running average root zone soil moisture (RZSM) falls below the 4-pentad (20 day) running average for a
period of at least 4 pentads; (2) by the end of the period, RZSM drops below the 20[th] percentile for that time of year according to
the 1979-2018 period of record. Figure 2 shows an example for the proposed definition applied over Montana, where the vertical
red-shaded region represents the suggested flash drought onset. However, specifying the duration of the event, including transition
from flash drought to standard drought, is a subject of ongoing research. RZSM is chosen over the surface SM on account of its
relevance to vegetation, low noise relative to surface soil moisture, and consistency with previous studies' recommendations (Ford
and Labosier, 2017; Hunt et al., 2009). Within the framework of the SMVI, the 1-pentad running average represents rapid changes
in RZSM (short memory), while the 4-pentad running average represents slower changes (longer memory). The 20[th] percentile
threshold is selected as recommended by the USDM to represent "Moderate Drought – D1" conditions, under which vegetation
may start showing signs of water stress. The minimum intensification period of 4 pentads is consistent with recommendations from
Otkin et al., ( 2018) that a 2-week period of rapid intensification is the minimum length required to capture rapid changes relevant
to vegetation health.
SMVI is a soil moisture-based index (yellow box in Fig. 1). The strength of the novel SMVI method lies in its ability to capture
rapid changes with respect to a slower drying trend. The index is sensitive to interruptions in drought onset, however, as it can be
reset by rain events. Since RZSM is key to computing SMVI—as it is to several other flash drought definitions—we prioritize use
of a high-quality soil moisture estimate. For this reason, we use the Soil MERGE (SMERGE) product. SMERGE is a hybrid daily
12.5 km resolution product generated by combining satellite observations from the European Space Agency Climate Change
Initiative and the North American Land Data Assimilation System-2 (NLDAS-2;  Xia et al., 2012a, 2012b) Noah Land Surface
Model output for RZSM averaged from 0-40 cm (Tobin et al., 2019). The SMERGE dataset has been evaluated against NDVI
products as well as in situ observations, indicating reliability for agricultural and ecological applications. For drought monitoring,





this product has the advantage of offering spatially and temporally complete RZSM estimates on an NLDAS-2 grid, while
incorporating additional satellite-derived information intended to improve these RZSM estimates.

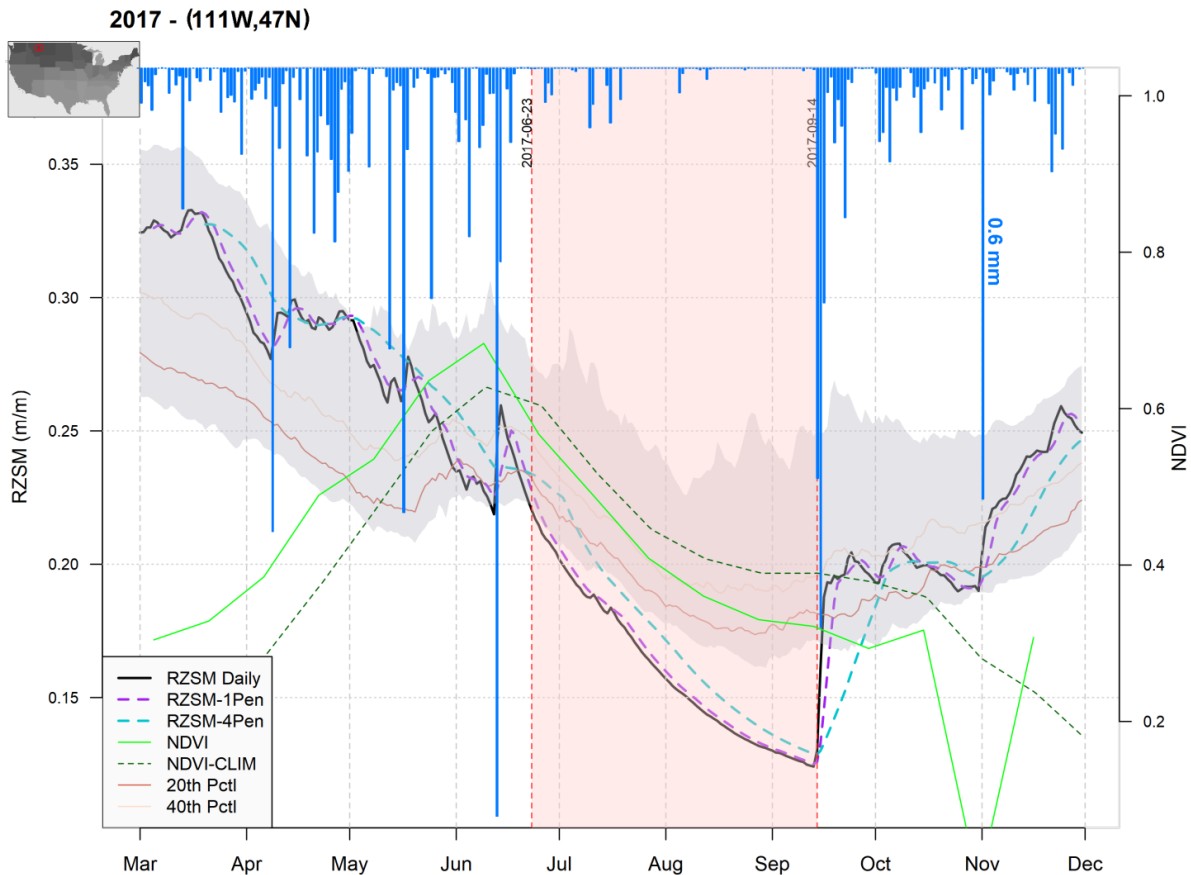

*Figure 2: SMVI proposed definition as applied to a grid point within the state of Montana in 2017. Shaded red region represents*
*the flash drought event. Gray shading represents the 10ᵗʰ to 90ᵗʰ percentile climatology of daily RZSM. Vertical blue bars are the*
*region's averaged daily precipitation.*
2.    SMPD (Soil Moisture Percentiles Drop)
Ford and Labosier (2017) introduced a definition based on a characterization of flash drought as a rapid descent into agricultural
drought conditions, referred to hereafter as the Soil Moisture Percentiles Drop (SMPD) method. It defines flash drought onset as
occurring when the 1-pentad running average RZSM falls from the 40ᵗʰ to the 20ᵗʰ percentile in a period less than or equal to 4
pentads. The original definition is based on RZSM from the NLDAS-2 (Xia et al., 2012a, 2012b) dataset in the eastern U.S. for
the top 40 cm of the soil column. Here, we apply the definition to gridded 12.5 km resolution SMERGE data for the 1979-2018
period to generate a dataset that can be compared to those derived using other definitions. Like SMVI, SMPD is a soil moisture-
based index (yellow box in Fig. 1).
3.    SESR (Standardized Evaporative Stress Ratio):
Whereas SMVI and SMPD focus directly on soil moisture, the Standardized Evaporative Stress Ratio (SESR) of Christian et al.,
(2019a) diagnoses flash drought occurrence on the basis of the normalized ratio between estimated actual and potential
evapotranspiration. This approach is guided by the principle that development of vegetation stress is key to an impactful flash
drought event, and this stress induces a rapid decrease in the transpiration flux during the drought intensification process (Basara
et al., 2019; Christian et al., 2019b, 2020). For SESR, six pentads (30 days) is defined as the minimum length for flash drought
development with a final SESR value less than the 20ᵗʰ percentile. These two criteria are used to satisfy the drought component of



flash drought and to capture flash drought events that lead to drought impacts. The rate of rapid drought intensification is evaluated
with two additional criteria. The first criterion requires a change in SESR between pentads  less than the 40th percentile This
criterion also allows for a temporary relaxation of the threshold for only one pentad to account for temporary mild weather
conditions or small rainfall events as long as the successive pentad does not exceed the 40th percentile(Christian et al., 2019a). The
second criterion requires the mean change in SESR be less than the 25th percentile to ensure that the events identified have an
overall rapid rate of development toward drought conditions. The more lenient 40th percentile threshold is used to account for large
variations in rapid drought development while still capturing periods with worsening environmental conditions. Together, the
pentad-to-pentad change threshold (40th percentile) and the mean rate of change (25th percentile) work in tandem to identify flash
drought events with rapid drought development. SESR has strong criteria that limit flash drought identification to very rapid
drought development, and so it is designed not to capture "flash drought" unless there are general drought conditions. Variables
used in SESR are shown in the cyan boxes in Fig. 1.
In this paper we use SESR exactly as it was implemented in the original publication, using the North American Regional Reanalysis
(NARR) dataset to provide input variables. NARR is a high-resolution atmospheric reanalysis for North America, performed at
approximately 0.3-degree resolution. The NARR is an appropriate dataset for hydrological applications due to the improved
analysis of the climate variability and diurnal cycle within the model and data assimilation system (Mesinger et al., 2006). We re-
grid SESR to match the 12.5-km resolution of the other products (SMERGE and NLDAS-2).

137        4.   HWD (Heatwave Driven):

In a set of papers, Mo and Lettenmaier (2015, 2016) introduce two paradigms for flash drought definitions. The first is a heatwave
driven (HWD) flash drought definition, which diagnoses flash drought conditions for any pentad in which the 2 m air temperature
anomaly is greater than one standard deviation, 1 m depth SM falls below the 40th percentile, and the evapotranspiration anomaly
is greater than zero. This third condition is designed to capture events in which high temperature and low soil moisture are defining
characteristics, but for which evapotranspiration has not yet become anomalously low. The HWD definition incorporates
information from the red, yellow, and (actual ET) cyan box in Fig. 1.
We apply the HWD definition using NLDAS-2 meteorological forcing data and the NLDAS-2 implementation of the Noah Land
Surface Model. We use NLDAS-2 because SMERGE does not contain all variables required for the calculation. However, we have
confirmed that replacing NLDAS-2 RZSM with SMERGE RZSM has little impact on our HWD flash drought inventory.

147        5.   PDD (Precipitation Deficit Driven)

The second paradigm suggested by Mo and Lettenmaier (2015, 2016) is the precipitation deficit driven flash drought (PDD). In
this study we have adopted their recommended definition where in a 1-pentad period precipitation drops below the 40th percentile
and the 2 m air temperature anomaly is greater than one standard deviation (similar to the HWD), while the evapotranspiration
anomaly is negative. The PDD definition incorporates information from the red, blue, and cyan boxes in Fig. 1. Like the HWD,
we have also used the NLDAS-2 forcing and Noah model datasets to calculate the definition and to inventory our results.
We note that PDD and HWD differ from other proposed flash drought indices in their explicit use of multiple meteorological and
hydrological variables. Additionally, these definitions diagnose flash droughts on the basis of the duration of anomalies rather than
their change over time. That is, flash droughts in PDD and HWD are acute deviations from climatology, rather than periods of
rapid intensification.

157        6.   USDM (U.S. Drought Monitor)

The United States Drought Monitor (USDM) (Svoboda et al., 2002), produced by the National Oceanic and Atmospheric
Administration, the United States Department of Agriculture, and the National Drought Mitigation Center, classifies drought into
5 intensity categories, ranging from Abnormally Dry (D0) to Exceptional Drought (D4). The USDM is produced in a hybrid
process, in which regional expert "authors" are provided information on more than 40 drought-relevant variables, and these authors
then work as a team to establish the drought map each week. The final product embodies a best estimate of drought conditions as
informed by quantitative indicators, field reports, and expert judgment. Data are released as shapefiles, which we rasterized to
match the resolution of the other products. Following Chen et al. (2019), we then define a flash drought as a degradation of two
categories or more in a four-week period. The USDM-based flash drought definition potentially includes all boxes in Fig. 1, as the
USDM authors are provided with information on all of these variables. USDM data are available from 2000-present.

167        7.   QuickDRI (Quick Drought Response Index)


QuickDRI (Quick Drought Response Index) is a Classification and Regression Trees (CART) machine learning model developed
by the National Drought Mitigation Center (NDMC) and the Center for Advanced Land Management Information Technologies
(CALMIT) at the University of Nebraska. The index was developed specifically to capture rapidly changing drought conditions.
QuickDRI maps drought intensification across CONUS at 1-km, weekly resolution on the basis of nine variables (two vegetation,
two hydrologic, one climatic, and four static biophysical parameters) to estimate drought conditions, with resulting drought
intensification values scaled according to the Standardized Precipitation Evaporation Index (SPEI) (https://quickdri.unl.edu/). The
QuickDRI inputs span the yellow (included as the soil moisture), blue (included as the standardized precipitation index - SPI),
cyan (included as the evaporative stress index - ESI), and green (included as the standardized vegetation index - SVI) boxes in Fig.
176  1.

As QuickDRI generates estimates of drought intensification as a continuous variable, it is necessary to define a threshold for flash
drought occurrence. We set this threshold as one standard deviation below the 4-week historical normal, referred to hereafter as
the QuickDRI model flash drought definition (QD1.0). Since QuickDRI relies heavily on real-time remotely sensed data, there are
gaps and noise in the record that must be addressed. We fill in missing data through linear temporal interpolation, and we mask
values greater than ± 4 standard deviations. QuickDRI data are available from 2000-present.
2.2.  Methods
The analyses presented here have been organized using Bukovsky Regions. The Bukovsky Regions are 29 eco-regions over United
States, Canada, and northern Mexico designed to represent climatically homogeneous areas. They are similar to the National
Ecological Observatory Network (NEON) (Kampe, 2010) ecological regions, with similar sensitivity to variations in regional
climatology (Bukovsky, 2011). Analyses were conducted over the 17 unique regions within CONUS (Fig. 3) as well as the eight
grouped regions as suggested by Bukovsky (2011). Here we present results for a subset of regions that capture a relevant diversity
of results, while results for all regions are available at **(**https://github.com/mosman01/Flash_Droughts/**).**
The flash drought inventories presented in this paper are based on flash drought occurrence: as soon as a flash drought is identified
according to a given definition in a given grid cell, that grid cell is tallied as having experienced flash drought in that year. That is,
we are concerned with spatial pattern and general seasonality of the occurrence of flash drought events as diagnosed by different
definitions. Intensity and duration of drought are not evaluated. Also, since definitions differ in if and how they mark the end of a
flash drought event, we count only the first flash drought identified for a grid cell in each year. The season of this flash drought
(MAM, JJA, or SON) is assigned based on onset date. This approach risks missing cases where two distinct flash drought events
hit a single location in one growing season, but it allows for a consistent inventory across definitions on the basis of "years with
flash droughts." The problem of counting multiple events at the same location in a single year using different definitions is a point
for further research, as differences and ambiguities in how different definitions define the end of a flash drought can lead to cases
where one definition diagnoses multiple flash droughts within a period that is classified as a single flash drought in another
definition. We do note that this approach captures the first drought, so it undercounts late season droughts if they occur in the same
location as an early season drought.  When calculating frequency, we use all the available data for each definition from 1979 to
201  2018.

For results presented by Bukovsky Region we calculate the percentage of area within each region hit by flash drought in each year.
This metric is used for qualitative comparison of definitions for selected events and for quantitative comparison using Pearson
correlations. Spearman and Kendall correlations were also calculated but yielded similar results and are not presented. Finally, an
analysis of the trends in flash droughts annual footprint is carried out for each climatic region within the Bukovsky regions using
the Mann-Kendell nonparametric trend test. Trend analysis is only performed for the definitions that can be calculated for the full
40-year period (1979-2018).





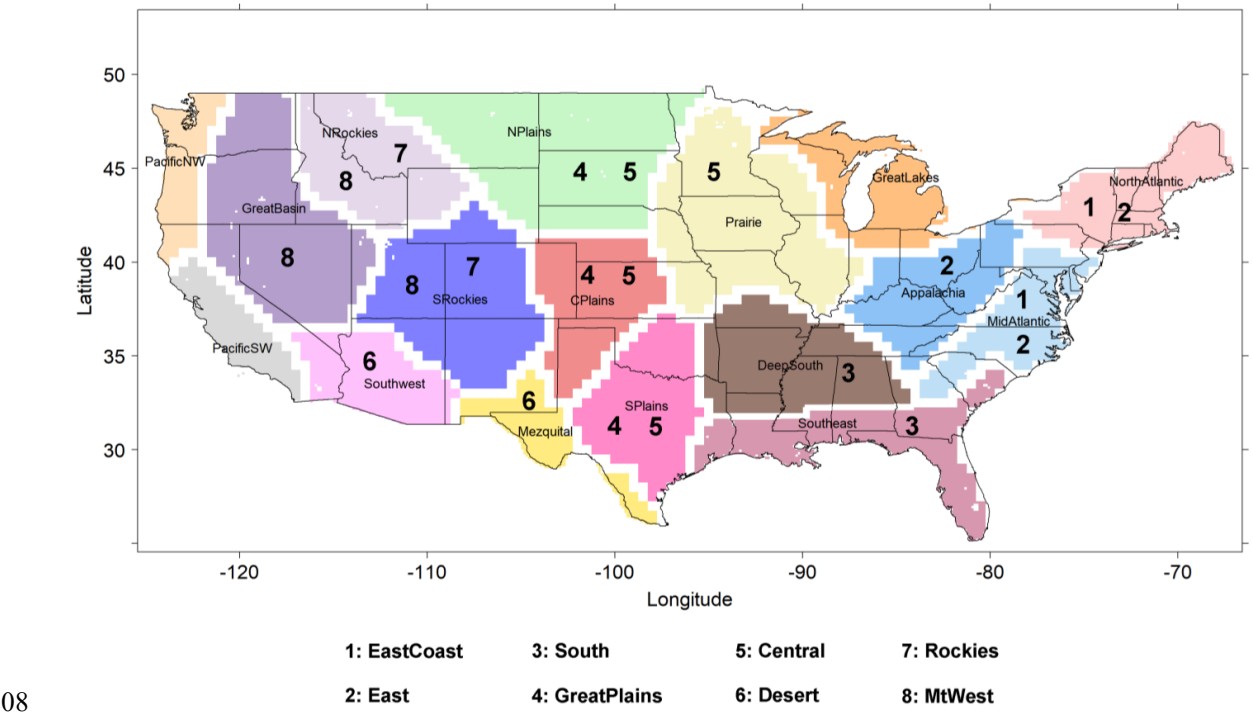

| | | | |
|---|---|---|---|
| 1: EastCoast | 3: South | 5: Central | 7: Rockies |
| 2: East | 4: GreatPlains | 6: Desert | 8: MtWest |


*Figure 3: Bukovsky regions within CONUS. Numbers represent groups of regions of similar climate characteristics.*



## 3. Results and Discussion:

### 3.1. Spatial distribution of flash droughts

As flash droughts have become recognized as a significant climate hazard, one key question is whether certain regions have an elevated probability of experiencing flash drought. As shown in Fig. 4, the seven drought definitions considered in this paper offer different answers to this question. This figure depicts the frequency of flash drought onset at each grid point within the specified season over the period of data availability for each definition through 2018. As noted in Christian *et al.* (2019a), the SESR identifies the Great Plains and western Great Lakes regions as hot spots for flash droughts. This band of high flash drought frequency running down the middle of the country resembles the region of strong land-atmosphere coupling identified in Koster *et al.* (2004) and in subsequent studies of climate feedback zones. In this sense, the SESR, which depends directly on the ratio of actual to potential evapotranspiration, may be emphasizing flash droughts that emerge through land-atmosphere temperature and evaporation couplings, which are strongest in transitional climate zones. There is a tendency for this SESR hot spot to emerge in the southern Great Plains in the spring (MAM) and to move further north in the summer (JJA).

Interestingly, this SESR pattern is nearly inverse to the pattern seen for PDD. In PDD, we see the strongest hotspot in the southwest, with a secondary maximum in the more humid eastern United States. While PDD includes actual evapotranspiration and temperature rules in its definition, it is designed to capture short meteorological droughts triggered by precipitation deficit. This results in higher frequencies in semi-arid regions with high precipitation variability and, to some extent, in regions where average rainfall is high and a significant negative anomaly in precipitation generally occurs in concert with the warm conditions required by the PDD definition. In contrast to PDD, the HWD yields a relatively uniform pattern of flash drought frequency, with lower totals overall.

Looking at the two soil moisture definitions, SMVI and SMPD, we see differences in overall frequency and spatial and seasonal distribution—which may reflect choice of threshold values. SMVI shows a relatively muted spatial pattern, with a broad maximum extending across the middle of the country and the western northern tier in summer, and a southwestern maximum in fall. SMPD has a springtime maximum in humid regions of the eastern United States and the Pacific Northwest, followed by a summertime pattern that includes significant frequency in the southwest. These differences trace to conceptual differences in the definition. Where SMPD focuses on soil moisture decline over several pentads, and thus is likely to capture vegetation-enhanced soil moisture draw-down that occurs in warm or dry springs in highly vegetated areas, SMVI controls for steady decline in order to isolate very rapid soil moisture drops. This makes it relatively less sensitive to seasonal forcing (e.g., warm springs leading to steady drying) and more sensitive to subseasonal processes. SMPD shows a noticeably high frequency of flash drought onset due to the duration threshold of 4 pentads or less, which allows short meteorological droughts to be misclassified as flash drought events.

Considering the hybrid products, USDM and QuickDRI both show a summertime maximum in flash drought frequency, but with distinctly different spatial patterns. In general, the QuickDRI areas of maximum frequency occur in drier regions in the western United States while USDM shows a maximum in the middle of the country that resembles the summertime SESR and SMVI patterns, though with a stronger maximum in Texas and Oklahoma. While it is difficult to diagnose the source of these patterns in a precise way given the composite nature of both products and the subjective component to USDM, it is likely that USDM authors are particularly attuned to agricultural impacts, and thus focus on rapid drying events that have severe impacts on crops and pastures, while the QuickDRI satellite-derived product may also be capturing variability in natural ecosystems and regions with less intensive agricultural activities. Different datasets and different algorithms involved within such complex model-based products could be a considerable source of uncertainty and variability.

The identification of geographic or seasonal flash drought hot spots, then, depends strongly on the definition. This choice of definition, in turn, will depend on the objective of the flash drought study. Investigating flash drought with an emphasis on vegetative impact, for example, might usefully apply a flux-informed definition like SESR, and would consequently focus on flash droughts in regions with land cover types associated with denser vegetation (e.g., agriculture, grasslands, and forests). A study or forecast system primarily concerned with the rapid intensification of a flash drought over either a humid or arid region might employ SMVI, which explicitly controls for more gradual drying in order to isolate the most rapidly intensifying portion of the events.


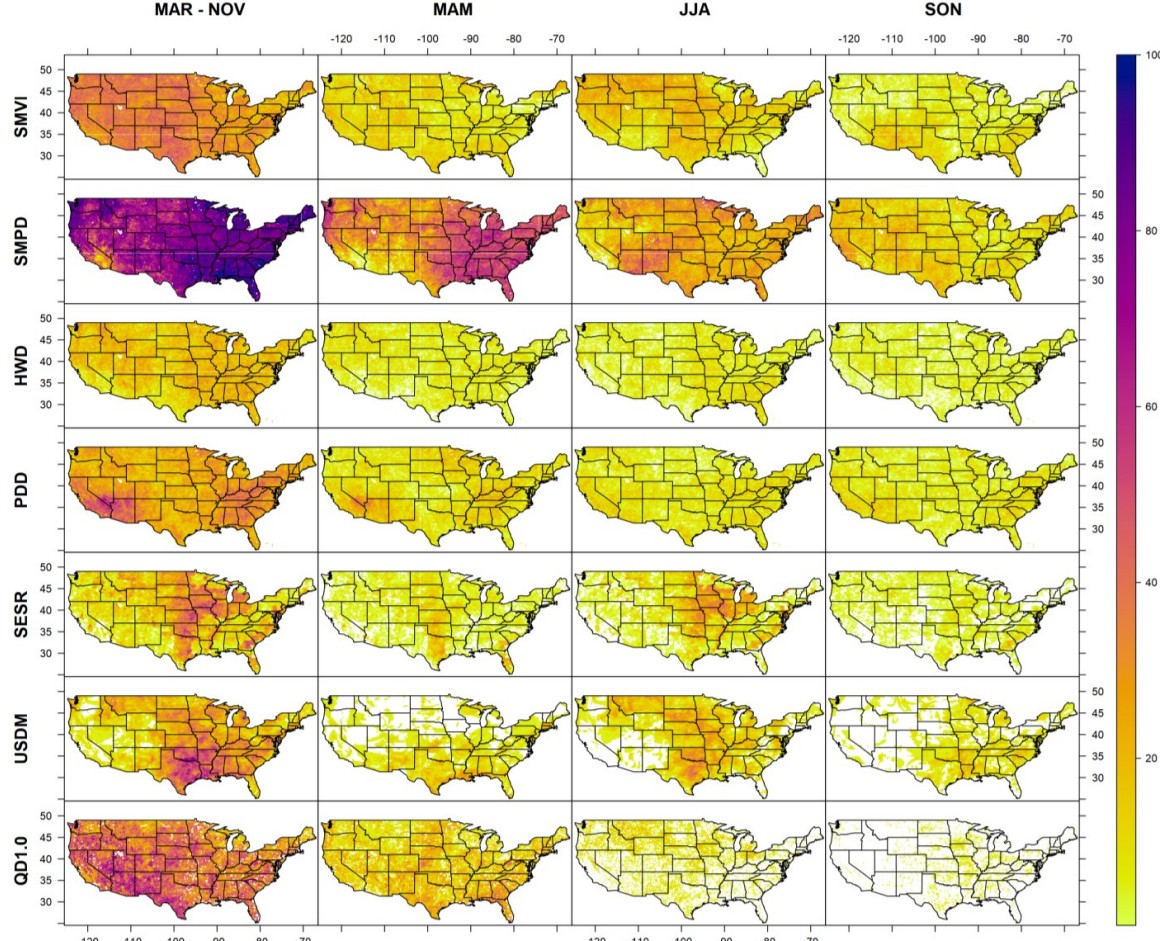

*Figure 4: Flash drought onset frequency for the selected definitions, calculated for the period of available data for each definition*
*through 2018 (1979-2018 for SMVI, SMPD, HWD, PDD, and SESR; 2000-2018 for USDM and QD1.0). White color represents*
*zero frequency.*
3.2.  Interannual variability
The definition-based differences in the geography and seasonality of flash drought frequency described above suggest that
definitions might also differ with respect to interannual variability. This is a particularly relevant issue for forecasting, as
differences in interannual variability imply differences in the prediction-relevant drivers of flash droughts. Indeed, if we examine
interannual variability in flash drought extent—defined as the percent area that experiences at least one flash drought in a given
year, within a specified region of interest—we see substantial differences between definitions. Figure 5 shows the Pearson's
correlation coefficients between different definitions' area hit by flash droughts annually for four different climatic regions. At
CONUS scale (Fig. 5a), the correlation between certain definitions, such as the two soil-moisture based definitions (SMPD and
SMVI) and the USDM is relatively high (> 0.7). This still leaves substantial unexplained variability between definitions, but the
differences between definitions are larger when comparing definitions that include other variables. SESR and PDD, for example,
have virtually no correlation in interannual variability at CONUS scale, which is consistent with the differences seen in Fig. 4 and
with the fact that the two definitions are based on very different principles and variables.
These differences become even more pronounced at regional scale. Figs 5b-d show regions in which differences are particularly
dramatic—the Southern Plains, Pacific Southwest, and North Atlantic Bukovsky Regions—and supplementary Fig. S1 shows the
remaining regions. We note that Fig. 5 is designed to highlight regions with substantial disagreement between definitions; the full
suite of regions shown in Fig. S1 includes a number of regions where definitions are in closer agreement with each other.





The Southern Plains is of particular interest, since it is a hotspot in the USDM-based definition and is an active agricultural region.
Here we see that the PDD and HWD definitions have no positive correlation with the USDM definition, which is again consistent
with differences in spatial patterns seen in Fig. 4 and with the fact that PDD and HWD are defined to capture short droughts rather
than periods of rapid intensification. Across other definitions, the correlations for the Southern Plains also tend to be (though are
not always) lower than the CONUS scale correlations. In the North Atlantic region, the PDD shows very weak correlations with
all definitions except the HWD since they share the common heatwave condition. Moving to the more arid Pacific Southwest and
Desert regions, we begin to see extremely low correlations across definitions, which in part reflects low signal to noise ratio for
drought indicators in dry climate zones and in part may point to implicit limitations in the useful climatic range of each definition.
In the Pacific Southwest, SESR stands out as having no positive correlation with any other definition, and the USDM also shows
very weak association with other definitions. This is a complicated region that includes arid zones and irrigated agriculture, which
would pose complications for an expert-informed composite indicator like USDM, and which is not represented in NARR or
NLDAS. Large expanses of arid areas with sparse vegetation coverage might also reduce the utility of a flash drought indicator
based on the actual to potential evapotranspiration ratio, such as SESR. Nevertheless, it is still possible that rapid onset droughts
matter in the region, particularly if they drive up irrigation demand or impact natural semi-arid ecosystems. Specifically, for the
Pacific Southwest region, all definitions show relatively less flash droughts frequency (SMVI, SMPD, USDM, SESR, and QD1.0;
local minimums in Fig. 4) except for PDD.

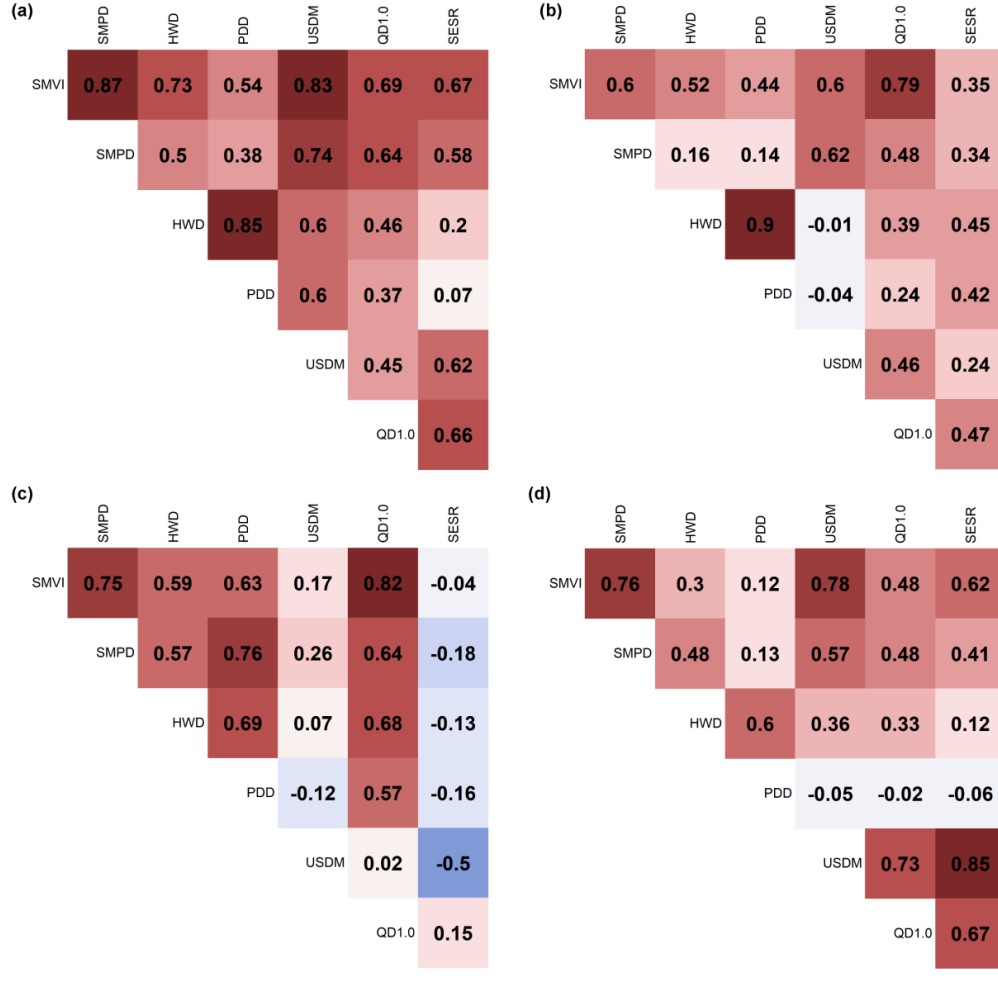


*Figure 5: Pearson's correlation coefficients matrix for the different definitions' percentage of area hit by flash droughts over the*
*Bukovsky regions: (a) CONUS, (b) Southern Plains, (c) Pacific Southwest, (d) North Atlantic.*





### 3.3. Representation of Major Flash Drought Events

Though there is no single agreed-upon definition for flash droughts, a number of major events in the past decade are widely recognized as having flash drought characteristics, to the point that these events can be thought of as canonical flash drought events. In addition, several major droughts that occurred prior to the popularization of the term "flash drought" have since been recognized as being consistent with flash drought. To obtain a clearer picture of how different definitions capture flash droughts, we examine several of these canonical flash droughts in greater detail.

We begin with an event that pre-dates the term flash drought, but has since been recognized as a member of the class (Basara et al., 2020; Jencso et al., 2019; Trenberth et al., 1988; Trenberth and Guillemot, 1996). The 1988 drought in the northwest, central and midwest United States developed over a period of less than 5 weeks, resulting in severe to extreme dry conditions over more than 10 states that cost the nation at least $30 billion dollars (National Oceanic and Atmospheric Administration, 1988). There was below average precipitation prior to the onset of the event, which contributed to its evolution. However, the most dramatic meteorological forcings were the pronounced and extended series of heatwaves that gripped the country in June, July, and August, and which were in their own right responsible for thousands of deaths (Changnon et al., 1996; Ramlow and Kuller, 1990; Whitman et al., 1997). These heatwaves occurred in combination with below average precipitation in June and July (Lyon and Dole, 1995). As this event predates QuickDRI and the USDM, we present a simple comparison of the other five flash drought definitions (Fig. 6). All definitions capture widespread drought, but timing and patterns differ. For example, whereas HWD emphasizes acute drought associated with high temperatures in JJA in the northern tier, SESR is more sensitive to evapotranspiration deficits across the middle of the country, which appear as a MAM signal in these seasonal maps. Similarly, SMPD is sensitive to dryness that appears in MAM, particularly in the eastern United States (consistent with the general spatial pattern of this definition; Fig. 6), while SMVI has characteristics of both the dry signal in the MAM window and intensification in the JJA period. We note that our seasonal cutoff dates are arbitrary, and could mask differences in timing within a season (e.g., March vs. May) while emphasizing relatively small timing differences that cross a seasonal break (e.g., May vs. June). Nevertheless, the analysis captures the general character of the seasonal timing of events.

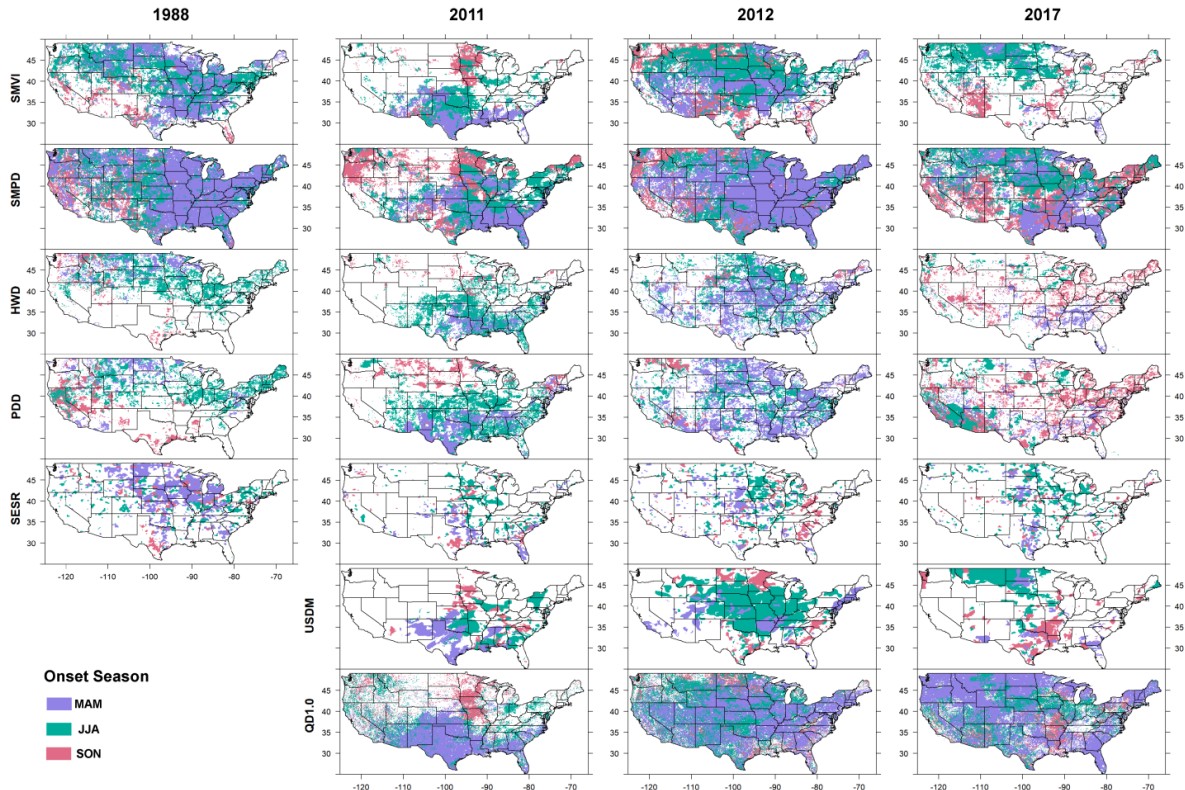

*Figure 6: Flash drought maps as captured by different definitions for the years 1988, 2011, 2012 and 2017. USDM and QD1.0 are available since 2000.*





Jumping forward, in 2011, the Southern Plains experienced a rapid onset, geographically focused flash drought that led into an
extended drought during the remainder of the year, making this one of the driest years in Texas since 1917 (Ejeta, 2012; Nielsen-
Gammon, 2012). The different flash drought definitions show signs of an early onset in spring in Texas and the southeast (Fig. 6),
which was the actual scenario according to the Office of the State Climatologist in Texas (Nielsen-Gammon 2012), that then spread
to other regions during the summer. SESR shows a more eastern pattern (where it is more humid), while the QD1.0 has a broad
drought signal across the southern tier of the county, but overall agreement across definitions is quite good. This suggests that the
2011 flash drought has a consistent signature in multiple meteorological and hydrological variables, which can be explained due
to the strong relationship between surface fluxes in the Southern Great Plains region (Mo and Lettenmaier, 2016).
The following year, 2012, produced one of the largest and most well documented flash droughts to date (Basara et al., 2019; Fuchs
et al., 2012; Hoerling et al., 2013, 2014; Mallya et al., 2013; Otkin et al., 2016). According to post-event analysis, large scale
teleconnections may have set the stage for the flash drought onset in spring and early summer (Basara et al., 2019; Fuchs et al.,
2012), with rapid intensification coming in summer as vegetation stress and heat set in. Results from the definitions (Fig. 6) show
different patterns for the spread of the drought. While an extensive drought in the middle of the country was in some form by all
definitions, the geographic pattern differed. Both HWD and SMVI, for example, capture a rapid drying in spring in Missouri and
surrounding regions, as abnormally warm conditions led to rapid soil moisture drawdown. The USDM-based definition, in contrast,
shows only limited drought in the MAM window, with widespread flash drought emerging in JJA. This likely reflects the fact that
the USDM did not make extensive use of vegetation indices in 2012, such that it is not optimized to capture rapid droughts (Senay
et al., 2008), and the warm spring conditions that set the stage for the catastrophic drought of summer are not identified as flash
drought when using the USDM as the input variable.
Finally, we examine the 2017 northern high plains flash drought. This was a geographically focused drought event that primarily
affected Montana, North Dakota, and South Dakota (Jencso et al., 2019). In contrast to the geographically focused flash drought
event of 2011, which was captured in a relatively similar way by most definitions, there is little consensus in the representation of
the 2017 event (Fig. 6). Both USDM and SMVI show spotty areas of drought in the northern high plains in MAM that expanded
during JJA. This pattern is almost entirely absent in HWD (despite the likelihood of being driven by reduction in snowpack due to
an early spring heat wave; Kimball *et al.*, 2019) and is evident only in spots in Montana for PDD and North Dakota for SESR.
SMPD identifies flash drought in this region in MAM and in some areas in JJA, but the region does not stand out relative to the
rest of the country. Similarly, QuickDRI shows widespread drought conditions that are not focused on the northern high plains.
These results show that the 2017 event qualified as a flash drought for some but not for all methods.

### 3.4. Climate drivers

Building on the event analysis presented in the preceding section, we now examine meteorological fields in the region of maximum
drought intensity for the 2011 and 2017 events—i.e., two regionally focused events, one of which presents relatively similar results
across all of the definitions (2011) and one which does not (2017). To simplify the problem, we examine only the main climate
variables used in creating the flash drought definitions (precipitation, RZSM, temperature, and actual and potential
evapotranspiration).
During the 2011 flash drought event, temperatures rapidly went extremely high and stayed that way for most of the spring and the
whole summer, as did potential evapotranspiration. While precipitation anomalies remained negative with very few exceptions,
actual evapotranspiration decreased just after the rapid increase in potential evapotranspiration. The RZSM shows a relatively rapid
decline in early summer, which occurs on top of a negative RZSM anomaly inherited from spring (Fig. 7a). In short, all of the key
variables applied in the flash drought definitions show a clear signal of rapid change to dry and hot conditions that were sustained
throughout the event, while precipitation stayed consistently low. For this type of event, choice of definition may not be critical
when attempting to characterize, monitor, or predict the drought.
In contrast, during the 2017 Northern High Plains drought (Fig. 7b) temperature was highly variable, and SM and ET did not fulfill
the HWD conditions for drought onset, so the HWD does not capture the observed drought onset. Precipitation was also less
consistent, explaining why PDD is spotty and may have missed the onset in multiple locations. Potential evapotranspiration,
interestingly, is fairly consistent even though temperature was noisy, so SESR captures the onset in some areas (though mostly
misses Montana), and RZSM gives the clearest signal, which is why SMVI and, to some extent, SMPD do well. In essence, the
2017 event is a flash drought primarily from the perspective of rapid soil drying, likely reinforced by high evaporative demand. It
is not a cleanly defined heatwave flash drought, and the rainfall signal is noisy. This suggests that efforts to understand and forecast
an event like 2017 will be concerned with different variables and different biophysical intensification processes than were active
in events like 2011.






*Figure 7: Timeseries of standardized main climate variables formulating the different flash droughts definitions averaged within regions of observed flash drought events. (a) 2011 flash drought observed over Southern Plains. (b) 2017 Northern Plains flash drought event. Grey horizontal lines represent ±0.5 standard deviation which is roughly equivalent to the 30th percentile of each variable's climatology.*



3.5. Trends
Over the past century there has been an increase in precipitation over much of the United States (IPCC, 2018). Studies over the
CONUS (Andreadis and Lettenmaier, 2006) also show positive trends in soil moisture and runoff, which lead to fewer hydrological
drought events. At the same time, temperature has increased for much of CONUS in recent decades, and Mo and Lettenmaier,
(2016) show that there was a dramatic increase in HWD events in the 90's due to this rapid warming. An increasing trend in flash
drought frequency according to this definition may be attributed to anthropogenic climate change as the rising temperature
increases evapotranspiration in humid and densely vegetated regions, which consequently causes a decrease in soil moisture (Wang
et al., 2016; Yuan et al., 2019).
In our analysis of flash droughts trends from 1979 to 2018 (USDM and QuickDRI definitions are not included due to the short
period of data availability), we see an increase in areas hit by HWD and PDD over most of the CONUS region in the past decade
(2009-2018) compared to (1979-1988) and almost no difference in SM and evaporative demand-based flash droughts definitions
(Fig. 8 and Fig. 9). Insomuch as HWD and PDD indices capture acute drought anomalies rather than the rapid intensification
targeted by other definitions, these results suggest that there is consensus across definitions that the frequency of rapidly
intensifying flash droughts did not increase between 1979-1988 and 2009-2018.
Considering each Bukovsky region (Fig. 8), however, we do observe different patterns of change in percentage of area experiencing
flash droughts over time. For example, the western coast (Pacific regions and Southwest) show an increase in areas experiencing
flash droughts while the Northern Plains and Rockies are characterized by a decrease in flash drought impacted areas. PDD shows
positive trends in almost all regions, and about half of the regions show a statistically significant trend. HWD is also positive in
almost all regions, with the majority of these trends showing statistical significance (Mann-Kendall test at $p < 0.05$; Fig. 8). Trends
in PDD and HWD are also positive and significant for CONUS on the whole. Trends for SMVI, SMPD, and SESR are mixed in
sign and generally not significant.
The presence of significant trends in PDD and HWD can be attributed to the fact that both directly depend, in part, on air
temperature. The other definitions are indirectly influenced by air temperature through its impact on evaporative demand and soil
moisture, but trends in those mediating variables are not as clear as the trend in temperature over the period of study. Insomuch as
there are systematic trends in flash drought across CONUS, then, it appears that those trends are only prevalent in definitions that
include the meteorological drivers of flash drought in the definition of the event. In this study, those definitions are limited to PDD
and HWD, which are definitions that target acute drought anomalies rather than rapidly intensifying flash droughts. The trends are
not evident when a definition depends only on a drought outcome of interest, such as soil moisture or evaporative stress. We do
note that there are very few cases of direct disagreement in sign between statistically significant trends across definitions. This
only occurs in the Central Plains, where SMPD differs in sign from HWD and PDD, and in the arid Great Basin region, where
SMVI shows a significant positive trend while SESR is significantly negative.






*Figure 8: Percentage change in areas hit by flash drought in (2009-2018) compared to (1979-1989) for CONUS and all Bukovsky*
*regions. Dashed black line represents the mean of all definitions per region. Significant trends (according to the Mann-Kendall*
*test) are marked by asterisks.*





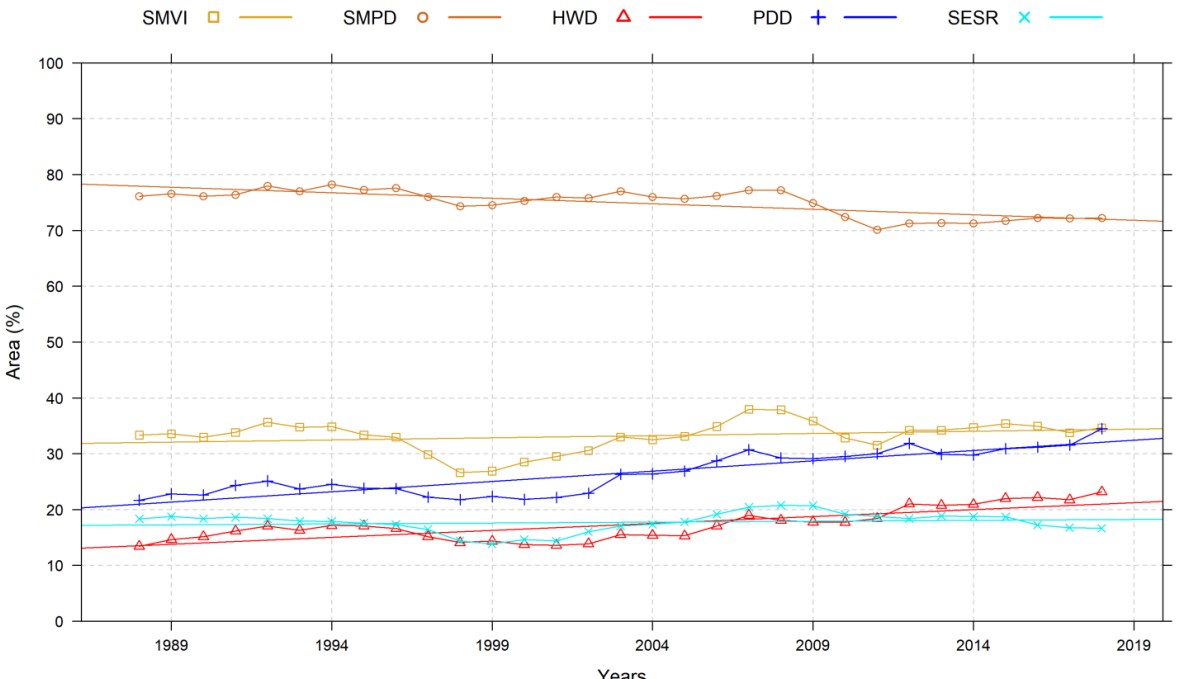

*Figure 9: 10-years running average percentage of area hit by flash droughts from March to November as estimated by different*
*definitions from (1979 to 2018). Linear trends are represented by the straight solid lines.*





**4. Conclusion:**
The present diversity in definitions of flash drought can be thought of as a feature, rather than a bug, of research in this field. This
diversity supports investigations of rapidly intensifying drought hazards from perspectives of meteorological forcing, drought
impacts, and various drought dynamics and feedbacks. However, trends and hotspots should be cautiously defined to avoid the
confusion that may arise due to the diversity of definitions and their ability to capture different aspects of flash drought. To answer
the question "are flash droughts increasing in the United States?" one needs to be clear on the manner in which the events are being
defined and calculated.
In applying definitions to the historic record, we see that the spatial coverage of some canonical flash drought events is well-
captured by most or all of the evaluated definitions. This includes the Southern Plains event of 2011, where consistent high
temperature and rainfall deficit led to a rapid and sustained increase in potential evapotranspiration, soil moisture drawdown, and
reduced evaporation. For other events, however, the definitions differed substantially in their assessment of the extent and timing
of the drought, or even on whether a notable flash drought had occurred at all. This was the case for the northern High Plains in
2017, for example, where variable temperatures and a noisy rainfall record interfered with some definitions, even as a rapid and
highly damaging drought struck the region. These results strongly indicate that "flash drought" represents a composite class of
events, with several possible pathways all leading to rapidly intensifying drought conditions. When assessing risk patterns,
developing forecast systems, or quantifying and projecting climate change impacts, it is critically important to be clear in the choice
of definition and the rationale in making that choice.
The SMVI definition shows an ability to capture the onset of major reported flash drought events regardless of the vegetation or
humidity conditions of the region. Ongoing research will enhance the definition's capabilities to report flash droughts severity and
intensity.
**Acknowledgement:**
This research was supported primarily by the National Science Foundation PREEVENTS program Grant number: 1854902. We
also would like to thank the research project team: Trevor Keenan from UC Berkeley, Christopher Hain and Thomas Holmes from
NASA and David Lorenz from University of Wisconsin-Madison for their helpful comments and discussion.
**Author Contribution:**
MO and BZ took the lead in writing the manuscript. BZ and HB supervised the formulation of the introduced definitions. JC and
TT provided research data and provided critical feedback and edits. JO and MA aided in interpreting the results and helped shape
the research and analysis. All authors discussed the results and contributed to the final manuscript.



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
