# Peer review of "Flash drought onset over the Contiguous United States: Sensitivity of"

_Hydrology and Earth System Sciences, 2020_

## Referee Comment (RC1) · Anonymous Referee #1 · 3 Sep 2020

Osman et al. provide an overview of definitions that have been developed to identify and quantify flash droughts (including a new definition developed by the authors here) and examine the robustness of these definitions with regard to characterizing flash droughts over the United States. They find that different definitions often lead to different conclusions with regard to flash drought frequency and trends, as well as the characterization of well-known past events. The results stress the importance of careful consideration of physical drivers when selecting a flash drought definition and the need to exercise caution when interpreting the results derived from a given definition. The paper is both informative and comprehensive and the topic is highly relevant to the broader scientific community, which is becoming more and more interested in the

topic of flash drought. I have only minor comments related to clarity and presentation, described below.

Comments:

Figure 1: While this figure is generally informative and understandable, some aspects of it are a bit confusing and can use further explanation/clarification. First, I suggest expanding the acronyms SM, PET, and AET on the figure. The general reader may not immediately know what these acronyms represent, especially since they do not appear to be defined prior to Fig. 1 in the text. Second, some aspects of the diagram itself are unclear. For example, it's unclear exactly what the box "pre-drought conditions" fundamentally represents and why there is an arrow drawn to it from "agriculture and ecological impacts" and another arrow drawn from this box toward "PET".Âă Also, it seems some information may be omitted from some boxes, which may raise questions – e.g., could PET itself also be a function of crop type and density, and isn't air temperature also a function of surface heat fluxes? Overall, I think it would be helpful to provide a brief explanation of this figure (either in the main text or as part of the caption), to clarify some of the issues raised above.Âă

L98 and in the Fig. 2 caption: Please define NDVI and briefly explain what this quantity represents. Including a reference that provides more information would also be helpful.

L122-126: The procedure to compute SESR could use more clarification. As currently written, the method is difficult to understand, particularly with regard to changes in SESR and how they relate to the given percentiles (40th, 25th).Âă A suggestion is to emphasize that the change in SESR must be less than the Nth percentile of SESR changes (determined from a distribution of SESR changes, with lower percentiles representing more negative changes or larger decreases).

L283: "SESR stands out as having no positive correlation with any other definition" - There is indeed one positive correlation. I suggest adding the phrase "(except with QD1.0, which is small)" to the end of this sentence.

Fig. 5 and especially Fig. S1: It would help to display the region name above each panel.

L339-340: Could you say a bit more about the scientific consensus on when the 2017 flash drought actually occurred, as done for the other 3 events? Is it believed to have started in the summer?

Fig. 9: Is this for CONUS? Please clarify in the figure caption.

Typos/writing:

Abstract, L17: "several types of event" -> "several types of events"

L62: "is the concept of flash drought robust to different definitions" should end with a question mark.

L289: I suggest changing "less flash droughts frequency" to "lower flash drought frequency"

Fig. 7: For temperature, the legend shows a square but on the plot it is an "x". Please correct.

---

## Referee Comment (RC2) · Anonymous Referee #2 · 30 Sep 2020

Major concerns:

>Though there are a bunch of flash drought definition, it is generally accepted by the scientific community that flash drought should emphasize the intensification rate to distinguish other types of drought [1]. I think the HWD definition is not suitable for flash drought, considering two aspects: 1) this definition cannot descript the rapid intensification of flash drought; 2) this definition may not be able to distinguish between flash droughts and short-term compound dry-hot events, leading to miscalculate flash droughts. Assuming that during dry-hot summer, conditions of HWD definition are relatively easy to meet, but actually such conditions may not form flash drought. Please

clarify how to distinguish between flash droughts and short-term compound dry-hot events in this paper.

Reference: [1] Otkin, J. A., Svoboda, M., Hunt, E. D., Ford, T. W., Anderson, M. C., Hain, C., Basara, J. B., Otkin, J. A., Svoboda, M., Hunt, E. D., Ford, T. W., Anderson, M. C., Hain, C. and Basara, J. B.: Flash Droughts: A Review and Assessment of the Challenges Imposed by Rapid-Onset Droughts in the United States, Bull. Am. Meteorol. Soc., 99(5), 911–919

>The presentation of typical flash drought events is weak and needs more specific cases. The authors may wish to show the temporal variation of real-world flash droughts in a Bukovsky Region, and further compare the differences of flash drought monitoring ability between definitions;.

>The climate variation during typical events should also be shown to help understand climate drivers, if climate data are availabe. In addition, in order to reflect whether these events have real impacts, it is better to analyze the changes of vegetation indicators (such as NDVI) , rather than just present description. Regarding these, I'm not very convinced that SMVI definition can well capture flash drought onset in both humid and arid regions.

>The authors shows the climate variation for typical regions during 2011 and 2017 flash droughts. I think it cannot well descript climate driver for the occurrence of flash drought, because such long-term climate anomalies could also lead to traditional droughts. I suggest that authors only focus on climate anomalies during flash drought events, such as extreme atmospheric anomalies (like rainfall defict, high surface temperatures, strong winds, or clear skies).

Other comments >Line 48: Please illustrate here that each color represents the flash drought definition. >Line 80: When the RZSM contains several layers, which layer of soil water should be selected? >Line 256: Please re-draw the Fig. 4. The legend can be a clear color segment. >Line 318: Figure 6 shows the frequency of flash drought

during typical years or the values of the indices? Please make it clear.

---

## Author Comment (AC1) · 24 Oct 2020

Osman et al. provide an overview of definitions that have been developed to identify and quantify flash droughts (including a new definition developed by the authors here) and examine the robustness of these definitions with regard to characterizing flash droughts over the United States. They find that different definitions often lead to different conclusions with regard to flash drought frequency and trends, as well as the

characterization of well-known past events. The results stress the importance of careful consideration of physical drivers when selecting a flash drought definition and the need to exercise caution when interpreting the results derived from a given definition. The paper is both informative and comprehensive and the topic is highly relevant to the broader scientific community, which is becoming more and more interested in the topic of flash drought. I have only minor comments related to clarity and presentation, described below.

»Reply: We would like to thank the reviewer for the supportive and constructive comments. The original comments and our responses are noted after each comment.

Comments:

Figure 1: While this figure is generally informative and understandable, some aspects of it are a bit confusing and can use further explanation/clarification. First, I suggest expanding the acronyms SM, PET, and AET on the figure. The general reader may not immediately know what these acronyms represent, especially since they do not appear to be defined prior to Fig. 1 in the text. Second, some aspects of the diagram itself are unclear. For example, it's unclear exactly what the box "pre-drought conditions" fundamentally represents and why there is an arrow drawn to it from "agriculture and ecological impacts" and another arrow drawn from this box toward "PET".Âa Also, it seems some information may be omitted from some boxes, which may raise questions – e.g., could PET itself also be a function of crop type and density, and isn't air temperature also a function of surface heat fluxes? Overall, I think it would be helpful to provide a brief explanation of this figure (either in the main text or as part of the caption), to clarify some of the issues raised above.Âa

»Reply: Thank you for these suggestions. Regarding presentation of the figure, we have expanded all acronyms as the reviewer suggests. Regarding substance, we have attempted to clarify what is meant by "pre-drought conditions" in the text (revised manuscript lines 34-37 and line 49, with reference to Wolf et al., 2016). The reviewer's

points about missing elements and simplifications in other components of the diagram are also appreciated. We recognize that there are many ways to think about different processes feedbacks and interactions with environment, and that our schematic does not capture all possible links and feedbacks. We now emphasize in the text (line 34) that the diagram is a simplification that shows "key" processes identified in previous literature. We do this to provide insight and a general framework for some interactions between environmental processes that could help in identifying pathways for the onset of a flash drought event.

L98 and in the Fig. 2 caption: Please define NDVI and briefly explain what this quantity represents. Including a reference that provides more information would also be helpful.

»Reply: Revised manuscript is updated with the clarification.

L122-126: The procedure to compute SESR could use more clarification. As currently written, the method is difficult to understand, particularly with regard to changes in SESR and how they relate to the given percentiles (40th, 25th).Âa A suggestion is to emphasize that the change in SESR must be less than the Nth percentile of SESR changes (determined from a distribution of SESR changes, with lower percentiles representing more negative changes or larger decreases).

»Reply: We appreciate that the SESR method may appear confusing due to the multiple criteria and thresholds that can be difficult to follow. To address the reviewer's specific point, the percentiles defined in the SESR method are based on the climatology of SESR at every grid point as defined by Christian et al. (2019a). In an effort to provide as much detail as possible within the constraints of the current manuscript, we have simplified the SESR description in section 2.1 in the revised manuscript to remove the confusion with percentiles used. For further details we refer the reader to Christian et al. (2019a), as it would take quite a significant amount of space to offer a full explanation and rationale for SESR methods.

L283: "SESR stands out as having no positive correlation with any other definition"

[Figure]

There is indeed one positive correlation. I suggest adding the phrase "(except with QD1.0, which is small)" to the end of this sentence.

»Reply: Agreed. Revised manuscript is updated with the suggestion.

Fig. 5 and especially Fig. S1: It would help to display the region name above each panel.

»Reply: Labels added to Fig. 5 and Fig. S1.

L339-340: Could you say a bit more about the scientific consensus on when the 2017 flash drought actually occurred, as done for the other 3 events? Is it believed to have started in the summer?

»Reply: The 2017 Flash drought had started with as small footprint in April and May and the onset then spread widely over the three impacted states. Text is updated with these information and references added (Line 371)

Fig. 9: Is this for CONUS? Please clarify in the figure caption.

»Reply: Yes. Revised manuscript is updated with the clarification.

Typos/writing:

Abstract, L17: "several types of event" -> "several types of events"

»Reply: Revised manuscript is updated with the correction.

L62: "is the concept of flash drought robust to different definitions" should end with a question mark.

»Reply: Revised manuscript is updated with the correction.

L289: I suggest changing "less flash droughts frequency" to "lower flash drought frequency"

»Reply: Revised manuscript is updated with the suggestion.

Fig. 7: For temperature, the legend shows a square but on the plot it is an "x". Please correct.

»Reply: Revised manuscript is updated with the correction.

» References:

Christian, J. I., Basara, J. B., Otkin, J. A., Hunt, E. D., Wakefield, R. A., Flanagan, P. X. and Xiao, X.: A Methodology for Flash Drought Identification: Application of Flash Drought Frequency Across the United States, J. Hydrometeorol., JHM-D-18-0198.1, doi:10.1175/JHM-D-18-0198.1, 2019a.

Wolf, S., Keenan, T. F., Fisher, J. B., Baldocchi, D. D., Desai, A. R., Richardson, A. D., Scott, R. L., Law, B. E., Litvak, M. E., Brunsell, N. A., Peters, W. and Van Der Laan-Luijkx, I. T.: Warm spring reduced carbon cycle impact of the 2012 US summer drought, Proc. Natl. Acad. Sci. U. S. A., 113(21), 5880–5885, doi:10.1073/pnas.1519620113, 2016.

---

## Author Comment (AC2) · 24 Oct 2020

»Reply: We would like to thank the reviewer for the supportive and constructive comments. The original comments and our responses are noted after each comment.

Major concerns:

>Though there are a bunch of flash drought definition, it is generally accepted by the scientific community that flash drought should emphasize the intensification rate to distinguish other types of drought [1]. I think the HWD definition is not suitable for flash drought, considering two aspects: 1) this definition cannot descript the rapid intensification of flash drought; 2) this definition may not be able to distinguish between flash droughts and short-term compound dry-hot events, leading to miscalculate flash droughts. Assuming that during dry-hot summer, conditions of HWD definition are relatively easy to meet, but actually such conditions may not form flash drought. Please clarify how to distinguish between flash droughts and short-term compound dry-hot events in this paper.

Reference: [1] Otkin, J. A., Svoboda, M., Hunt, E. D., Ford, T. W., Anderson, M. C., Hain, C., Basara, J. B., Otkin, J. A., Svoboda, M., Hunt, E. D., Ford, T. W., Anderson, M. C., Hain, C. and Basara, J. B.: Flash Droughts: A Review and Assessment of the Challenges Imposed by Rapid-Onset Droughts in the United States, Bull. Am. Meteorol. Soc., 99(5), 911–919

»Reply: First, we thank the reviewer for clarifying the aspects required to define flash droughts and highlighting some deficiencies that may arise within definitions such as HWD. We agree with the reviewer that the HWD definition is not necessarily a definition suitable for capturing flash drought events as it does not count for rapid intensification and it is only based on anomalies within a short period. Both HWD and PDD are introduced in this paper since they are widely used in flash droughts identification literature despite their major limitation. The presented comparison emphasizes the limitation within these definitions in a fair and objective matter. We have clarified these points in multiple sections within the manuscript: Lines 161-169, Lines 291-292, Lines 365-366, Lines 391-392 and Lines 416-418.

>The presentation of typical flash drought events is weak and needs more specific cases. The authors may wish to show the temporal variation of real-world flash droughts in a Bukovsky Region, and further compare the differences of flash drought monitoring ability between definitions;.

[Figure]

»Reply: We thank the reviewer for pointing out the need for more specific case studies and clarification. The revised manuscript is updated with more of the suggested discussion. The main purpose of the paper is to compare different flash drought definitions and explore how different criteria – with careful selection - may be applied to define flash droughts in the context of proposed mechanisms of interest. The four case studies are intended to provide examples that may be familiar to readers and that can make the conceptual distinctions between definitions more concrete. We do not attempt fully detailed case analysis of these events. As the reviewer suggests, we do make use of Bukovsky Regions to understand variation of real-world flash droughts, though we do this for flash drought statistics rather than for time series analysis of the case study events. In section 3.1 we discuss the differences between the different definitions in terms frequency of occurrence and spatial differences and the possible reasons for these variations. Bukovsky regions are presented in more detail in the next sections as we look into the correlations, interannual variability, trends and climate drivers. Section 3.3 discusses the onset and conditions of the observed 1988, 2011, 2012 and 2017 flash droughts (2016 flash drought is added to the revised manuscript; Lines 333-335, Lines 355-366) and highlights the spatial and temporal differences in capturing flash droughts' onset between the different definitions. Figure 7 shows time series of variables relevant to different flash drought definitions for the selected case studies, and the associated text (Section 3.4) describes the relevance of those time series to the drought monitoring ability of different definitions.

>The climate variation during typical events should also be shown to help understand climate drivers, if climate data are availabe. In addition, in order to reflect whether these events have real impacts, it is better to analyze the changes of vegetation indicators (such as NDVI) , rather than just present description. Regarding these, I'm not very convinced that SMVI definition can well capture flash drought onset in both humid and arid regions.

»Reply: Thanks to the reviewer for emphasizing the importance of showing the vegetation impact to support the introduced definition. We agree that NDVI is a powerful vegetation impact indicator. We have included it in a descriptive way, as the reviewer notes; e.g, Figure 2 (Line 107) depicts an example for the SMVI definition for a selected grid point within the state of Montana in 2017 and shows how the NDVI drops below the climatological mean values for the same grid point. We do not pursue a full quantitative analysis of vegetation indicators of drought in this manuscript, in part because these analyses require careful consideration of metrics, timing, and ecological context that would require substantial expansion of the paper. We intend to undertake such analyses in future papers. In order to offer better vegetation context for the events analyzed in this manuscript, we have added Figures S2 and S3 (shown below as Fig.1 and Fig.2 respectively) to the revised manuscript to illustrate the tempo-spatial change in NDVI within selected flash drought impacted regions in 2012 and 2017 respectively. The change in NDVI anomalies show similar patterns and timing to these captured by SMVI. Regarding SMVI, In Section 3.3 we present examples for major flash droughts (1988, 2011, 2012, 2016 and 2017) that span a range of climatic regions. The SMVI definition appears to perform well across these diverse regions. For example, 1988 historical flash drought hit many parts of the US covering humid regions (such as the Great lakes region) and semi-arid regions (such as Northern plains). SMVI successfully captured the event as observed (Lines 324-331), and did so again for the climatically extensive 2012 flash drought (Lines 347-354). The 2011 flash drought is another example for which SMVI captures an event that includes semi-arid regions, this time in Texas. That said, we acknowledge the reviewer's implication that arid zones are not fully explored, and that vegetation might not respond to flash droughts in a truly arid region in a manner that would demonstrate SMVI performance. For this reason, we have replaced "arid" with "semi-arid" in all passages that refer to SMVI performance.

>The authors shows the climate variation for typical regions during 2011 and 2017 flash droughts. I think it cannot well descript climate driver for the occurrence of flash drought, because such long-term climate anomalies could also lead to traditional droughts. I suggest that authors only focus on climate anomalies during flash drought

events, such as extreme atmospheric anomalies (like rainfall defict, high surface temperatures, strong winds, or clear skies)..

»Reply: Thank you for the constructive suggestion and underlining the importance of focusing on climate anomalies during flash droughts. Assuming that the comment about section 3.4, the presented analyses show only the standardized anomalies for the main variables involved within the discussed definitions during the onset year only. The discussion is focused on the onset season as observed and calculated. In lines 383-389, we explain the observed climate conditions (in terms of anomalies) during the 2011 flash drought that show early signs of drought intensification during spring and remain for the summer before recovering in fall. In lines 390-395, the discussion is focused on the climate conditions as illustrated in Fig. 7b and how some climate variables may not be appropriate to use for identifying flash droughts; for example, depending on temperature anomalies only would lead to mischaracterization of the event, or even missing it completely, as happened for the HWD and (partially) the PDD definitions.

Other comments:

»Reply: Many thanks to the reviewer for the comments and suggestions.

>Line 48: Please illustrate here that each color represents the flash drought definition.

»Reply: Revised manuscript is updated to clarify that colors are used to represent the different definitions (Lines 109-110).

>Line 80: When the RZSM contains several layers, which layer of soil water should be selected?

»Reply: SMERGE dataset used contains RZSM of the 0-40cm layer. However, if the average of multiple layers from a different dataset is used, similar results would be achieved since the power of the SMVI definition is the relative comparison between two moving averages. Line 98 clarifies the confusion.

>Line 256: Please re-draw the Fig. 4. The legend can be a clear color segment.

»Reply: Thank you. Figure updated in the revised manuscript as suggested.

>Line 318: Figure 6 shows the frequency of flash drought during typical years or the values of the indices? Please make it clear.

»Reply: Figure 6 shows the onset of major flash drought events in the different discussed years (section 3.3) marked by seasons. Caption is edited in the revised manuscript for clarification.

**NDVI Anomaly ( % )**

USDA · NASA

GIMMS Global
Agricultural Monitoring

**Terra MODIS 8-day**

| | |
|---|---|
| −125 | Below |
| −80 | Normal |
| −60 | |
| −40 | |
| −30 | |
| −20 | |
| −10 | |
| −5 | |
| 5 | Normal |
| 10 | |
| 20 | |
| 30 | |
| 40 | |
| 60 | |
| 80 | Above |
| 125 | Normal |

No Data
Water

04/14/2012 - 04/21/2012

06/17/2012 - 06/24/2012

07/19/2012 - 07/26/2012

05/16/2012 - 05/23/2012

**Fig. 1.** Tempo-spatial change in NDVI within selected flash drought impacted regions in 2012.

NDVI Anomaly ( % )

| | |
|---|---|
| -125 | Below |
| -80 | Normal |
| -60 | |
| -40 | |
| -30 | |
| -20 | |
| -10 | |
| -5 | Normal |
| 5 | |
| 10 | |
| 20 | |
| 30 | |
| 40 | |
| 60 | |
| 80 | Above |
| 125 | Normal |

No Data
Water

USDA
NASA

GIMMS Global
Agricultural Monitoring

Terra MODIS 8-day

06/02/2017 - 06/09/2017

05/01/2017 - 05/08/2017

07/04/2017 - 07/11/2017

05/17/2017 - 05/24/2017

**Fig. 2.** Similar to Fig. S2 for 2017 flash drought.

---

## Author Response (AR2)

**Flash drought onset over the Contiguous United States: Sensitivity of inventories and trends to quantitative definitions**

Mahmoud Osman[1], Benjamin F. Zaitchik[1], Hamada S. Badr[1], Jordan I. Christian[2], Tsegaye Tadesse[3],

Jason A. Otkin[4], Martha C. Anderson[5]

**Author's response**

We would like to thank the reviewers for the supportive and constructive comments. The original comments are noted in black and our responses are noted after each comment are in blue.

**Anonymous Referee #1**

The authors have adequately addressed my previous comments and I feel the paper is now ready for publication. I just recommend a few grammatical corrections to a new paragraph about the 2016 flash drought:

L342: change "southeast has been hit" to "southeast was hit"

Fixed in the revised manuscript as suggested.

L342: change "which has sparked" to "which sparked"

Fixed in the revised manuscript as suggested.

L347-349: I suggest rewriting this sentence as "The 2016 flash drought was expected to extend eastward towards the Carolinas, but heavy precipitation from the tropical storms and hurricanes (Hermine and Matthew) that hit the region ended the catastrophic event (Konrad II and Knox, 2018)."

Fixed in the revised manuscript as suggested.

L351: change "SESR definition has underestimated" to "The SESR definition underestimated"

Fixed in the revised manuscript as suggested.

**Anonymous Referee #2**

>Section 3.1 shows spatial distribution of flash droughts, I can see the spatial difference due to different definitions, however, I'm not sure which definition is better for application. The aim of this study is to evaluate the current diversity of flash drought definitions, so I think the contents in section 3.1 is not enough. I would prefer a specific presentation of a typical flash drought event than detailed case analysis of these events . Figure 2 well shows that the SMVI can capture flash drought, I suggest that the authors can add the analyses for this grid cell using other definition, which could illustrate the difference using the different definition. In addition, the variation in climate factors during this event can be illustrated.

We thank the reviewer for pointing out that the analysis presented in section 3.1 is not consistent with the aim of evaluating flash droughts against one another. While we indicate areas of agreement and disagreement between definitions and with the drought monitor, we do not demonstrate that one definition is "better for application." However, this is by design. The aim of the paper is to present the diversity of flash drought definitions and to compare in a manner that addresses the research questions stated in lines 65-70. We do not intend to rank the definitions based on the power for applications. Rather, we feel that the diversity of flash drought inventories presented by different definitions reflects the fact that the definitions are capturing different types of events; there may not be a "typical" flash drought across all credible definitions (see, for example our text on lines 53-57). Rather than restructure Section 3.1 into an argument for a single, best definition, then, we have clarified our intent by adding text to the final paragraph of the introduction (line 70): "We emphasize that the comparison of definitions is not designed to choose a single, "best" way to define flash droughts. Rather, cases of divergence between definitions can be used to examine different characteristics of rapidly intensifying drought events."

For the second part of the comment: *"Figure 2 well shows that the SMVI can capture flash drought, I suggest that the authors can add the analyses for this grid cell using other definition, which could illustrate the difference using the different definition. In addition, the variation in climate factors during this event can be illustrated."* We appreciate these suggestions. To address the question regarding how different definitions behaved at this specific grid cell, we have added a marker to indicate the location of the grid cell in the 2017 panels of Figure 6. This allows the reader to see how other definitions captured the flash drought at this location. For the climate factors, we note that Figure 7, bottom panel, presents the requested climate analysis, albeit over an average area rather than this specific grid point. The results for the single grid point have been added as Supplementary Figure S1, and is now referenced in line 89.

>Line 87, "Figure 2 shows an example for the proposed definition applied over Montana", Please revise this illustration, as Figure 2 only relies on a grid point within Montana.

The caption was updated to include the suggested edit in the first revised version of the manuscript.

>I think a full quantitative analysis of the response of NDVI to flash drought is not required, while showing the NDVI change during a flash drought can help to illustrate whether the definition can capture flash drought. Regarding this, I would like to suggest that the authors can add more sub temporal panels in Figure S2-3. In addition, the spatial extent (longitude and latitude) for sub figures need to be added.

Thanks for the suggestion. NDVI supplementary figures (S3 and S4) are now redrawn for better quality and updated with extra snapshot from the available MODIS NDVI anomaly data to show more temporal variation.

>Some of the figures are not clear enough to read, please improve them.

Thank you for noting that. All figures are generated in high quality, but this has most probably occurred in the reviewer's manuscript version due to conversion to pdf format. We will make sure to upload the original quality figures for publishing.